# The Role of Tetrapyrrole- and GUN1-Dependent Signaling on Chloroplast Biogenesis

**DOI:** 10.3390/plants10020196

**Published:** 2021-01-21

**Authors:** Takayuki Shimizu, Tatsuru Masuda

**Affiliations:** Graduate School of Arts and Sciences, The University of Tokyo, 3-8-1 Komaba, Meguro-ku, Tokyo 153-8902, Japan; ctshimizu@g.ecc.u-tokyo.ac.jp

**Keywords:** plastid, tetrapyrrole, heme, chloroplast, retrograde signaling, *gun* mutants

## Abstract

Chloroplast biogenesis requires the coordinated expression of the chloroplast and nuclear genomes, which is achieved by communication between the developing chloroplasts and the nucleus. Signals emitted from the plastids, so-called retrograde signals, control nuclear gene expression depending on plastid development and functionality. Genetic analysis of this pathway identified a set of mutants defective in retrograde signaling and designated *genomes uncoupled* (*gun*) mutants. Subsequent research has pointed to a significant role of tetrapyrrole biosynthesis in retrograde signaling. Meanwhile, the molecular functions of GUN1, the proposed integrator of multiple retrograde signals, have not been identified yet. However, based on the interactions of GUN1, some working hypotheses have been proposed. Interestingly, GUN1 contributes to important biological processes, including plastid protein homeostasis, through transcription, translation, and protein import. Furthermore, the interactions of GUN1 with tetrapyrroles and their biosynthetic enzymes have been revealed. This review focuses on our current understanding of the function of tetrapyrrole retrograde signaling on chloroplast biogenesis.

## 1. Introduction

Tetrapyrroles are involved in various functions critical to whole organisms’ viability, including light absorption, electron transfer, and oxygen binding [1,2]. Thus, they are essential components of primary metabolism, such as respiration and photosynthesis. Tetrapyrroles contain four pyrroles, aromatic rings containing four carbon atoms and one nitrogen atom, in linear (e.g., bilins) or cyclic (e.g., porphyrins) chemical structures. Porphyrins often chelate central metal ions, such as Co^2+^, Fe^2+^ or Fe^3+^, or Mg^2+^ ions. Meanwhile, fully conjugated (pigmented) porphyrin rings possess photodynamic properties: they can generate reactive oxygen species (ROS), primarily singlet oxygen under light excitation, which cause photooxidative damage and cell death [3]. Therefore, organisms must strictly regulate tetrapyrrole biosynthesis. In plants and algae, tetrapyrroles’ main end products are siroheme, heme, phytochromobilin, chlorophyll (Chl) *a,* and Chl *b*. Although they are synthesized in plastids, these tetrapyrroles are widely distributed, with the exception of Chls. Especially, heme is found throughout the cell. In addition to prosthetic groups’ function, tetrapyrroles have been proposed as signaling molecules that control transcription and intracellular signaling. This review focuses on the signaling function of tetrapyrroles on chloroplast biogenesis in *Arabidopsis thaliana*. We inform interested readers of several comprehensive reviews on the signaling function of tetrapyrroles on other aspects of plant physiology [4,5,6].

## 2. Biosynthesis of Tetrapyrroles in Plants

### 2.1. The C_5_ Pathway and the Common Pathway

In plant cells, tetrapyrrole biosynthesis takes place entirely in the plastid (Figure 1). The first committed precursor for all tetrapyrroles is 5-aminolevulinic acid (ALA). In plants, algae, and many bacteria, ALA is synthesized from glutamate via the C_5_ pathway [7]. In this pathway, glutamate is first ligated with plastid-encoded tRNA^Glu^ to form glutamyl-tRNA^Glu^, a substrate for plastid protein biosynthesis. The following two enzymes, glutamyl-tRNA reductase (GluTR) and glutamate 1-semialdehyde aminotransferase (GSAT), synthesize ALA from glutamyl-tRNA^Glu^. In particular, the step of GluTR is the rate-limiting step of total tetrapyrrole biosynthesis, the activity of which is controlled by transcriptional and post-transcriptional regulations [8,9]. *Arabidopsis* has two paralogous genes for GluTR, *HEMA1* and *HEMA2*. *HEMA1* is light-responsive, is actively expressed in green tissues, and contributes predominantly to Chl biosynthesis [10,11]. ALA dehydratase condenses two molecules of ALA onto the monopyrrole to form porphobilinogen (PBG). PBG deaminase assembles four PBG molecules, which are further assembled onto the tetrapyrrole precursor uroporphyrinogen III (Urogen III). Three stepwise oxidation steps oxidatively decarboxylate Urogen III and the final step enzyme, protoporphyrinogen IX oxidase, oxidizes the colorless protoporphyrinogen IX to fully conjugated and pigmented protoporphyrin IX (Proto). Alternatively, Urogen III can be directed to siroheme biosynthesis. Since the pathway from ALA to Proto is conserved among most organisms, this pathway is called ‘the common pathway’.

### 2.2. Chl Branch and Chl Cycle

The next branchpoint involves the insertion of either Mg^2+^ or Fe^2+^ by Mg-chelatase (MgCh) or ferrochelatase (FC), respectively, directing Proto into the Chl or heme biosynthetic pathways. MgCh consists of three subunits, CHLI, CHLD, and CHLH in plants. In *Arabidopsis*, CHLD and CHLH are encoded by a single gene, and CHLI is encoded by two isoforms, *CHLI1* and *CHLI2*. CHLI1 is essential for photosynthesis [12,13], whereas CHLI2 has a minor role in assembling the MgCh complex [14]. Additionally, GUN4 enhances MgCh activity by mediating substrate or product channeling [15,16,17]. In the Chl branch, MgCh catalyzes the formation of Mg-protoporphyrin IX (MgProto), which is methylated by *S*-adenosylmethionine MgProto methyltransferase (encoded by *CHLM*) to form MgProto methyl ester (MgProtoME). MgProtoME cyclase catalyzes the formation of the fifth ring of the tetrapyrrole ring structure, which is further converted to 3,8-divinyl protochlorophyllide *a* (DV-Pchlide *a*). DV-Pchlide *a* is further converted to Pchlide *a* by DV-Pchlide *a* 8-vinyl reductase. Pchlide *a* accumulates in dark-grown angiosperms because the next enzyme, light-dependent NADPH:Pchlide oxidoreductase (POR), requires light to reduce Pchlide *a* to chlorophyllide *a* (Chlide *a*). Depending on the plant species, it is considered that the step of DV-Pchlide *a* 8-vinyl reductase occurs after POR reaction (Figure 1) [9]. Then, Chlide *a* is esterified with a geranylgeraniol or phytol by Chl synthase to form Chl *a*, some of which is reversibly converted to Chl *b* via the Chl cycle [8].

### 2.3. Heme Branch

In the heme branch, FC inserts Fe^2+^ into Proto to produce protoheme (heme *b*), which is the prosthetic group of *b*-type cytochromes and proteins, such as catalase and peroxidase. In these hemoproteins, the heme is noncovalently bound via coordination to the Fe atom by histidine and/or cysteine residues [18]. There are two isoforms of FC (*FC1* and *FC2*) in *Arabidopsis* and cucumber, which show differential tissue-specific and development-dependent expression profiles: *FC2* is light-dependent and mainly expressed in photosynthetic tissues, whereas *FC1* is stress responsive and ubiquitously expressed in all tissues [19,20]. Some protoheme is further metabolized into other hemes, such as heme *a* and heme *c.* Protoheme is also substrate for bilins. Heme oxygenase oxidatively cleaves protoheme to biliverdin IX. Then, phytochromobilin synthase converts biliverdin IX to *3Z*-phytochromobilin. Finally, *3Z*-phytochromobilin is isomerized to *3E*-phytochromobilin, which functions as the chromophore for the phytochromes (PHYs) [21].

## 3. Coupling of Two Genomes Is Required for Chloroplast Biogenesis

In plant cells, the chloroplast is one of the differentiated states in which plastids have a photosynthetic function [22]. In the meristem of angiosperms, plastids exist as undifferentiated proplastids, and chloroplasts can directly form from the proplastids with developmental cues and light signals. This process is called chloroplast biogenesis. During chloroplast biogenesis, thylakoids are formed and stacked into defined grana. The thylakoids are the internal lipid membranes interlaced with protein complexes, which provide the platform for the light reactions of photosynthesis [22,23]. In the absence of light, proplastids differentiate into etioplasts with unique lattice membrane structures called prolamellar bodies (PLBs), which accumulate Pchlide *a* and POR. Once the etiolated seedlings are exposed to light, most Pchlide *a* molecules in PLBs are immediately converted to Chlide *a* by POR, and then to Chl *a* via enzymatic processes [24].

Plastids originate from a free-living cyanobacterium in a process known as endosymbiosis [25]. A primitive cyanobacterium was engulfed by a non-photosynthetic eukaryotic cell and coexisted in ancient times. Many genes of the cyanobacterium endosymbiont are thought to be lost or transferred to the nucleus of the host cell following endosymbiosis. Despite this, some genes involved in photosynthesis, transcription, and translation were retained in plastid genomic DNA [26,27,28]. Photosynthesis in chloroplasts is a reaction that uses light in the photochemical system at the level of thylakoids. The carbon fixation system (Calvin cycle) present in the soluble stroma fraction. Since the protein complexes responsible for these two reaction systems are composed of proteins encoded by nuclear and chloroplast genes, coordinated gene expression between the two components is necessary for functional chloroplast biogenesis.

Thus, for efficient chloroplast biogenesis, communication between the nucleus and the plastids is paramount. The nucleus controls most aspects of chloroplast biogenesis (“anterograde signaling”) [29,30], while plastids are also thought to emit signals that alter nuclear gene expression (“retrograde signaling”). So far, multiple signaling pathways have been proposed to be plastid-to-nucleus communication. In general, the retrograde signals are categorized into two classes: (i) “biogenic control” signals that mainly act during the initial stage of chloroplast development, and (ii) “operational control” signals that are primarily generated in response to environmental stimuli in matured chloroplasts [31]. For evaluation of the biogenic control, the relationship between chloroplast function and nuclear gene expression at the initial stage of seedling development has been mainly evaluated. Meanwhile, the operational control is occurred in matured chloroplasts. This control is proposed to include three chloroplast redox signals: (i) the redox states of components of the photosynthetic electron transport (PET) chain, primarily plastoquinone, (ii) redox-active thiol group-containing proteins and antioxidants couples to PET, and (iii) the generation of ROS [31]. As this review focuses the biogenic control, so interested readers are encouraged to see several comprehensive reviews about the operational control [32,33,34].

Important insights into biogenic control of retrograde signals have come from the finding that the expression of many *photosynthesis-associated nuclear gene*s (*PhANG*s) is dependent on the presence of functional chloroplasts [35,36]. The perturbation of chloroplast function by mutations or treatments with inhibitors results in the strong down-regulation of many *PhANG*s [37]. Subsequently, a set of mutants, called *genomes uncoupled* (*gun*) mutants, which have a reduced ability to coordinate this nuclear response to the chloroplast function, were identified through the retention of *PhANG*s, such as *Lhcb* gene expression after treatment with norflurazon (NF) [37]. So far, two major categories of mutants have been identified: mutants affected in tetrapyrrole metabolism [15,37,38,39] and mutants in the light signaling components [40].

## 4. Identification of *Gun* Mutants

The original *gun* mutant screening isolated five mutants (*gun1* to *gun5*) that retained the expression of *PhANGs* after NF treatment [37]. *gun2, gun3, gun4,* and *gun5* are the four mutants of tetrapyrrole biosynthetic genes and encode heme oxygenase, phytochromobilin synthase, and the regulator and the CHLH subunit of MgCh, respectively [38] (Figure 1). These results suggest the involvement of tetrapyrrole metabolism in biogenic retrograde signaling.

As discussed below, researchers of the retrograde signaling field related to *gun* mutants have struggled with some of the proposed signals and components of the signaling pathway. These discrepancies may be caused by phenotypic analysis of mutants and transgenic lines involved in retrograde signaling mainly via molecular genetic approaches. These approaches were: knockout or knockdown mutants or transgenic lines of *Arabidopsis* seedlings, developmental and growth (light intensity and sugar concentration) conditions, type and concentration of inhibitors used, and sensitivity of detection methods (RNA gel blot or quantitative reverse transcription-polymerase chain reaction (qRT-PCR)) in the *gun* phenotype evaluation (derepression of *PhANG* expression). In *gun* mutant screening, NF, an inhibitor of the carotenoid biosynthesis enzyme phytoene desaturase, is mainly used to block chloroplast functions that result in the intense repression of many *PhANG*s. Inhibition of carotenoid biosynthesis by NF may cause photooxidative stress during the conversion of proplastids to chloroplasts [35,36]. In general, *Arabidopsis* seedlings are grown on agar plates containing 1–5 µM NF for 4–10 days under illumination (~100 µmol m^−2^ s^−1^) for scoring of the *gun* phenotype. It is assumed that during growth on the NF-containing plates, free Chl or its precursors accumulate without the concomitant accumulation of carotenoids. In such a situation, ROS (singlet oxygen) are generated, which cause the photooxidative block of chloroplast biogenesis [35,36]. However, it is not conclusive whether singlet oxygen is generated transiently or consistently, or whether this ROS is directly involved in photooxidative bleaching by NF [6,41]. It is presumed that NF’s inhibitory mechanism on chloroplast biogenesis may be complex, making it difficult to evaluate the phenotype [42,43]. Light intensity also affects the ability of NF to repress *PhANG* expression [42]. Furthermore, as there is no clear threshold for determining the *gun* phenotype. It is sometimes difficult to distinguish whether tested lines are real *gun* mutants or not, and if the changes in *PhANG* expression are marginal or rare, but significant.

### 4.1. The Function of MgProto as a Negative Mobile Signal

The *gun4* and *gun5* mutations directly affect MgCh activity, and *gun2* and *gun3* mutants are unable to metabolize heme that may cause feedback inhibition of GluTR on ALA synthesis. These results led the hypothesis that the first intermediates of the Chl branch, MgProto and/or MgProtoME, function as mobile retrograde signals between the chloroplast and the nucleus. The signaling role of MgProto has been suggested in algae [44]. In the green alga *Chlamydomonas reinhardtii*, exogenous treatment of MgProto or MgProtoME induced the nuclear heat shock protein 70 (*HSP70A*) gene [45]. Additionally, in the red alga *Cyanidioschyzon merolae*, MgProto is proposed to function as a coordinator of the cell cycle from plastid-to-nuclear DNA replication [46].

In *Arabidopsis*, the *chld* [47,48] and *chli1/chli2* double mutant [48] were also shown to exhibit a *gun* phenotype. Furthermore, higher accumulation of MgProto after NF treatment in the wild type than in the *gun2* and *gun5* mutants was detected [47]. A treatment with MgProto but not protoheme, Proto, or PBG repressed *Lhcb1* expression in leaf protoplasts [47]. Subsequently, using confocal fluorescent microscopy, an accumulation of MgProto in the cytosol was observed in NF- and ALA-fed *Arabidopsis* seedlings [49]. These observations led to the proposal that MgProto functions as a negative mobile signal emitted from the chloroplast to repress *PhANG* expression [47].

Opponents have argued that *Arabidopsis* mutants deficient in the MgProto methyltransferase (*chlm*) [50] and MgProtoME cyclase (*chl27*) [49] accumulate high levels of MgProto, but neither mutant exhibits a *gun* phenotype. In barley, *LHCB* expression was greatly reduced in NF-treated seedlings, but no accumulation of MgProto was detected [51]. Furthermore, a detailed quantification of the tetrapyrrole intermediates in NF-treated *Arabidopsis* did not support any relationship between the levels of MgProto and *gun* phenotype in several of the *gun* mutants [52,53].

Meanwhile, a transient increase in MgProto that might regulate nuclear gene expression has been proposed [54,55,56]. In 3-week-old *Arabidopsis* seedlings, the levels of MgProto and MgProtoME increased, peaking 72 h after NF treatment—the profile of which is opposite to that of *Lhcb* expression in the wild type [56]. Using methyl viologen (MV) as an inhibitor, accumulation of MgProto and MgProtoME was reported in *Arabidopsis* after 3.5 h of treatment [54,55]. Contrastingly, using the dexamethasone-inducible RNAi system, *CHLH/GUN5*, *CHLM*, and *CHL27* were repressed in 10-day-old *Arabidopsis* seedlings, which caused a transient increase in the levels of MgProto within 24 h [57]. However, such a temporary increase in MgProto did not affect the expression of *PhANG*s, suggesting photooxidative damage is necessary to exhibit a *gun* phenotype. Concerning the cytosolic receptor of MgProto, the proteomic analysis identified heat shock proteins, especially in HEAT SHOCK PROTEIN 90 (HSP90), as MgProto-binding proteins [54,55]. In RNAi lines of *HSP90* genes in the *gun5* background, significantly decreased levels of derepression of *PhANG* expression were observed when compared to *gun5* in NF- or MV-treated seedlings [54,55].

It is currently unknown how MgProto and/or MgProtoME can be exported from the dysfunctional chloroplast to the cytosol. Since MgProto and MgProtoME contain a fully conjugated ring structure, which has photodynamic properties, deregulated accumulation of these intermediates may cause phototoxicity to the cell [4]. In addition, these unstable intermediates can be rapidly degraded under illumination. Considering derepression of *PhANG*s is observed in the *gun* mutants after several days on NF-containing plates when the levels of MgProto are quite low, other mechanisms than MgProto signaling are likely to be involved.

### 4.2. Function of Heme as a Positive Mobile Signal

An alternative hypothesis is that another metabolite, protoheme (hereafter just “heme”), functions as a positive signal. Heme has been proposed to be a regulatory factor in controlling transcription and intercellular signaling in yeast, animals [58,59], and algae [60,61,62]. In *Ch. reinhardtii*, heme is proposed as a signaling molecule that may substitute for light [60], and the expression of hundreds of genes was affected by exogenous heme treatment. However, only a few of them have been associated with photosynthesis [61]. In *Cy. merolae*, abscisic acid (ABA) induced heme-scavenging tryptophan-rich sensory protein-related protein (TSPO), resulting in inhibition of the cell cycle G1/S transition [62]. Since the addition of exogenous heme canceled the ABA-dependent inhibition of DNA replication, ABA and heme are assumed to have a regulatory role in algal cell cycle initiation [63]. It is noted that a homolog of TSPO in *Arabidopsis* showed heme-binding properties and was induced by ABA treatment [64]. However, *Arabidopsis* TSPO localized to the secretory pathway [64].

A dominant *gun* mutation (*gun6-1D*) resulting in the overexpression of *FC1* has led to a model in which FC1-derived heme mediates plastid signaling [39]. Although the *FC1*-overexpressing line (*gun6-1D*) exhibited the *gun* phenotype, the *FC2*-overexpressing line did not exhibit *gun* phenotypes, suggesting increased flux of the FC1-derived heme may act as a signaling molecule that controls the *PhANG*s [39]. In plants, the main FC activity is detected in chloroplasts and negligible activity is observed in mitochondria [65,66], although the possibility of mitochondrial localization of FC cannot be excluded [67]. In tobacco, the overexpression of *FC1* resulted in the detection of FC1 protein in mitochondria with a concomitant increase in mitochondrial FC activity [67]. A functional analysis of FC1 and FC2 has suggested that FC1 is key to providing non-photosynthetic heme required for extraplastidic organelles, while FC2 produces photosynthetic heme [68]. Partial compensation of *fc1* and *fc2* by the *FC2* and *FC1* genes, respectively, confirmed distinct functions of these FC isoforms [69]. The overexpression of *FC1* in plastids but not in mitochondria resulted in the *gun* phenotype, supporting the role of FC1-derived heme as a plastid-derived retrograde signal [70]. However, endogenous levels of the total [39,42] and free [71] heme did not correlate to a *gun* phenotype. Since FC1 and FC2 are colocalized to the plastid, the importance of clarifying the precise localization of FC1 and intracellular heme trafficking mechanism is emphasized [39]. Proteomic analysis of heme-binding proteins has identified several novel extraplastidic proteins, including nuclear-localized transcription factors, histone deacetylases, and RNA helicases from *Arabidopsis* and *Cy. merolae* [72]. Further clarification of these putative heme-binding proteins may be necessary for elucidating the heme signaling pathway.

### 4.3. Function of Other Signaling Components

ABSCISIC ACIC INSENSITIVE 4 (ABI4) is assumed to be the transcription factor that mediates retrograde signaling [73]. Actually, ABI4 is indeed featured prominently in published models [33,34,74,75]. However, this model is challenged by independent observations that *abi4* does not show a *gun* phenotype [76,77,78,79]. It is also proposed that PHD TYPE TRANSCRIPTION FACTOR WITH TRANSMEMBRANE DOMAINS (PTM) mediates retrograde signaling [80]. However, careful analysis has shown no significant involvement of PTM in retrograde signaling [81]. From these results, it is recommended to omit PTM and ABI4 in the model of biogenic retrograde signaling [76,77,78,79,81,82].

Contrastingly, it looks promising that biogenic retrograde signaling is closely linked to light signaling [78,83]. Screens for a *gun* mutant phenotype identified multiple alleles of the blue light photoreceptor cryptochrome 1 (*CRY1*). They also suggested a role for the red-light photoreceptor PhyB and the transcription factor ELONGATED HYPOCOTYL 5 (HY5) [40]. HY5 is one of the potent transcription factors that functions downstream of photoreceptors [84]. In the dark, HY5 is ubiquitinated and degraded by CONSTITUTIVE PHOTOMORPHOGENIC 1 (COP1), a ubiquitin E3 ligase that regulates the abundance of various light-signaling components in association with DE-ETIOLATED1 (DET1) [85]. When chloroplast biogenesis was blocked, CRY1 became a negative regulator of *Lhcb1* expression, because HY5 was converted from a positive to a negative regulator [40]. Meanwhile, *gun1 cry1* and *gun1 hy5* synergistically attenuated the plastid regulation of *PhANG* expression and chloroplast biogenesis, consistent with the integration of light and plastid-to-nucleus signaling [40].

GOLDEN2-LIKE (GLK) also contributes to the retrograde signaling. Many plant species have *GLK* genes in pairs (*GLK1* and *GLK2*). In *Arabidopsis*, *GLK1* and *GLK2* are functionally equivalent, and only the double knockout mutant (*glk1 glk2*) showed perturbed chloroplast development [86]. GLK1/2 is a key transcriptional regulator of photomorphogenesis that positively regulates the expression of a large number of *PhANG*s during chloroplast biogenesis [86,87,88] and also a major nuclear regulator of the retrograde signal [89]. GLK1 is a direct target of PHYTOCHROME-INTERACTING FACTOR 4 (PIF4) [90], one of the repressors of photomorphogenesis regulated by Phy [84]. Currently, it is not known whether GLK1 is targeted by PIF1 and PIF3, which are the main repressors of chloroplast development. During photomorphogenesis, Phy-mediated degradation of PIFs releases the repression of GLK1/2, promoting *PhANG* expression [78]. It is demonstrated that key tetrapyrrole biosynthetic genes are co-expressed with key nuclear-encoded photosynthetic genes [91,92]. Significant conservation of the HY5-binding G-box and GLK-binding motif (CCAATC) was found in the promoter region the of co-expressed genes [93]. Based on the observation of chloroplast development in *Arabidopsis* roots, it has been proposed that a combination of HY5 and GLK1/2 is crucial to the coordinated expression of *PhANG*s and key tetrapyrrole genes [94]. As the matter of fact, the overexpression of GLK1/2 caused a *gun* phenotype [88,95]. When chloroplasts were dysfunctional by oxidative stress, GLK1/2 expression was repressed in a GUN1-dependent manner, antagonizing the phytochrome signal and attenuating photomorphogenesis [78].

## 5. The Function of GUN1

Unlike mutant lines *gun 2* to *6* related to the tetrapyrrole biosynthetic pathway, *gun1* encodes a chloroplast protein containing a pentatricopeptide repeat (PPR) protein with a C-terminal small MutS-related (SMR) domain. Since *gun1* can also prevent down-regulation of *PhANG* expression after treatment with lincomycin (Lin), an inhibitor of plastid translation [73], GUN1 has been suggested to act independently of the tetrapyrrole-mediated GUN signaling pathway. Interestingly, *gun1* was shown to be hypersensitive to Lin or NF [96,97,98]. Since the PPR [99] and SMR [100] domains are known to be involved in nucleotide-binding, it was first suggested that GUN1 acts as a nucleotide-binding protein involved in plastid gene expression (PGE), plastid DNA metabolism, or DNA repair [73]. Subsequent efforts to screen for GUN1-associated partners by co-immunoprecipitation and mass spectrometry analysis identified many proteins rather than nucleotides [101,102,103]. The highly disordered domain at the N-terminus of GUN1 [104] may correspond to an intrinsically disordered region (IDR) [105]. The binding of protein partners induces a conversion of this domain to an ordered structure, which allows the same polypeptide sequence to undertake different interactions with different consequences. Nearly 300 different proteins involved in diverse biological processes in chloroplast were immunoprecipitated after crosslinking of GUN1–GFP in *Arabidopsis*, suggesting the promiscuous nature of the GUN1 protein [101]. Although the specificities to GUN1 were not identified, these putative GUN1-associated proteins were involved in transcription [97], translation [101,106], and import [107], all of which include homeostasis of chloroplast proteins [108,109,110] (Figure 2). In addition, enzymes involved in tetrapyrrole biosynthesis have been identified by yeast two-hybrid and bimolecular fluorescence complementation (BiFC) assays [101,107] (Figure 2).

Although *GUN1* is highly and consistently expressed, the protein levels of GUN1 remain not abundant because of its very high turnover [103]. The GUN1 protein was only detectable where active chloroplast biogenesis occurs, such as in cotyledons and leaf primordia initially after germination [103]. The rapid turnover of GUN1 is controlled mainly by the chaperone ClpC1, suggesting degradation of GUN1 by the Clp protease [103]. Inhibition of plastid translation by Lin or oxidative stress by NF may prevent the ClpC-dependent degradation of GUN1, resulting in higher accumulation of this protein under these conditions [103]. As GUN1 accumulates only at the very early stage of leaf development under natural conditions, it has been suggested to function in chloroplast biogenesis [103]. However, the function of GUN1 at later developmental stages has also been suggested [101,103]. Since overexpression of GUN1 caused an early flowering phenotype, it is hypothesized that GUN1 functions in developmental phase transitions beyond chloroplast biogenesis [103].

Concerning the localization of GUN1 in the plastid, it was first suggested that GUN1 localizes in nucleoids where plastid DNA is actively transcribed. Transiently expressed GUN1–GFP in tobacco (*Nicotiana benthamiana*) exhibited granular fluorescence colocalizing with pTAC2, a component of transcriptionally-active complexes [73]. Such fluorescence in GUN1 was also observed in the stable *Arabidopsis* GUN1–GFP line [101] and BiFC assays of GUN1 and its binding proteins [97,101]. Meanwhile, GUN1–GFP was detected in the stroma as a dispersed signal in the stably transformed *Arabidopsis* lines [103,107]. It was recently reported that GUN1 alters its sub-chloroplast localization after NF treatment [111]: a speckled pattern of fluorescence was detected in the untreated condition, while a diffused distribution was observed after NF treatment. Therefore, it is likely that such a different distribution of GUN1 may be caused by employed developmental stage or functionality of GUN1.

### 5.1. Function of GUN1 on Transcription and Editing of Plastid Genes

In chloroplasts, two different RNA polymerases are present to transcribe the chloroplast genes [112,113]: the nuclear-encoded polymerase (NEP), a monomeric T3-T7 bacteriophage-type enzyme, that is mainly responsible for transcription of housekeeping genes, and the plastid-encoded polymerase (PEP), a multimeric bacterial-type enzyme, that mainly transcribes photosynthesis-related genes. Chloroplast development is associated with a shift in the primary RNA polymerase from NEP to PEP.

In Lin-treated *Arabidopsis* seedlings or mutants with defective plastid protein homeostasis, the increase in NEP-dependent transcripts, such as *rpoA* and *rps12-3’*, was observed in the wild type, but was compromised in the *gun1* [97] mutant. GUN1 physically interacted with RpoTp encoding NEP and enhanced its activity upon depletion of PEP [97].

Additionally, GUN1 was proposed to interact with the MULTIPLE ORGANELLAR RNA EDITING FACTOR 2 (MORF2), a member of the so-called plastid RNA editosome, and regulate plastid RNA editing [114,115]. Compared with the wild type, the *gun1* mutant showed differential efficiency of RNA editing levels of 11 sites in the plastid transcriptome after NF or Lin treatment [114]. Target genes were NEP-dependent, including transcripts of the PEP core subunits. The editing sites correspond to highly conserved residues, suggesting a lack of GUN1 leads to the synthesis of an impaired form of PEP core proteins [114,115].

### 5.2. The Function of GUN1 on Translation of Plastid Genome

GUN1 interacts with several ribosomal subunits, such as the plastid-encoded ribosomal proteins S1 (PRPS1) and the nucleus-encoded plastid ribosomal protein L10 [101]. The *gun1* mutation genetically interacts with the mutations of these genes. Analysis of *gun1 prps1* lines indicates that GUN1 controls PRPS1 accumulation at the protein level [101]. Moreover, functional overlapping of GUN1 with RH50 encoding the plastid DEAD-box RNA helicase, which is a 23S−4.5S rRNA maturation factor, has been reported [116]. This suggests the involvement of GUN1 in plastid ribosome assembly. Furthermore, the interaction of GUN1 with FUG1/cpIF2 encoding the chloroplast translation initiation factor IF-2 was detected [101]. The *gun1* mutation aggravated the effects of decreased FUG1 levels on chloroplast protein translation [106]. Based on these results, the authors proposed that GUN1 is a modulator of plastid protein homeostasis, whose function only clearly manifests when plastid protein homeostasis is perturbed [106].

### 5.3. The Function of GUN1 on Protein Import into the Plastid

GUN1 has been proposed to be involved in the regulation of protein import into the plastid [89,97,107], although a critical point is remained to be clarified. GUN1 was shown to interact with the chloroplast chaperone cpHSC70-1 to promote the import of nuclear-encoded chloroplast proteins [101,107]. In addition, GUN1 was suggested to interact with other proteins involved in protein import, protein folding, and protein unfolding/degradation [101,108]. In *gun1*, the reduced accumulation of NEP-dependent transcription of *Tic214* [97], together with the diminished activity of cpHSC70-1 [107], likely leads to an import defect, resulting in the over-accumulation of precursor proteins in the cytosol [97,107]. *cphsc70-1* showed a *gun* phenotype in NF-treated, but not in Lin-treated, seedlings. The accumulation of precursor proteins in *gun1* was confirmed by independent analysis, which accommodates the higher cytosolic HSP90 and HSP70 accumulation [97]. The activity of HSP90 was positively correlated with *PhANG* expression and proposed to be directly involved in the development of the *gun* phenotype in the *gun1* mutant [107]. Since cytosolic HSP90 was identified as one of the MgProto-binding proteins and *hsp90* mutants showed reduced derepression of the *gun* phenotype [54,55], HSP90 has been proposed as the central cytosolic transducer of plastid retrograde signal [107] mediating the activation of a positive regulator of transcription such as HY5 [40] and GLK1/2 [88,89].

One critical point of this model is that a *gun* phenotype is not seen for other mutants with reduced NEP transcription, such as *sca3* defective in RpoTp enzyme, [97] or chloroplast import, such as the plastid protein import mutant 1 (*ppi1*) encoding *TOC33*, *toc75-III-3*, and *tic40-4* [107]. Proteomic analysis of the plastid protein import mutant 2 (*ppi2*) defective in *Toc159* revealed the accumulation of several plastid pre-proteins in the cytosol with concomitant upregulation of HSP90.1 protein, but *PhANG* expression is largely repressed in this mutant [89,117]. Therefore, it is likely that the GUN1-dependent modulation of import activity does not play a significant role in retrograde signaling. In addition, it is not clear why failure to import proteins into damaged chloroplasts should induce more expression of these pre-proteins, which may cause a catastrophic positive feedback loop.

### 5.4. The Link between GUN1 and Tetrapyrrole Biosynthesis

At first, a synergistic enhancement of the *gun* phenotype was observed in the *gun1-1 gun4-1* and the *gun1-1 gun5* double mutants relative to the single mutants [38], indicating the tetrapyrrole and GUN1 signals act independently. Subsequently, the *gun* phenotype of a double mutant *gun5 gun1-9*, a nonsense allele of GUN1, was found to be indistinguishable from *gun1-9,* suggesting the tetrapyrrole signal acts upstream of GUN1 [73]. Transcriptome analyses confirmed significant interactions between GUN5- and GUN1-dependent plastid signaling mechanisms [73]. The increased levels of ALA, heme, and Chl in *gun1 sig2* relative to *sig2* supported the interaction between these two plastid-to-nucleus signaling mechanisms [118]. A direct interaction between GUN1 and tetrapyrrole biosynthetic enzymes was revealed by immunoprecipitation of GFP-tagged GUN1 from the GUN1–GFP overexpressing line after treating chloroplasts with a crosslinking agent [101]. Yeast two hybrid and BiFC experiments confirmed the interaction of GUN1 with the CHLD subunit of MgCh, PBG deaminase, Urogen III decarboxylase, and FC1 [101] (Figure 1 and Figure 2). It is interesting to note that, with the exception of *CHLD*, the other three genes are not light responsive, but involved in the common pathway and the heme branch [91]. It should also be noted that interaction of GUN1 with tetrapyrrole biosynthetic enzymes are detected in independent immunoprecipitation analysis [107].

It was found that when ALA is fed to etiolated *Arabidopsis* seedlings, *gun1* accumulated more Pchlide *a* in darkness, while GUN1 overexpressors accumulated less Pchlide *a* when compared with the wild type [104]. Such higher Pchlide *a* accumulation in *gun1* was also observed without ALA feeding [119]. Since total heme levels were similarly changed in mutants and overexpressors, it was proposed that GUN1 controls the total tetrapyrrole flow [104]. Furthermore, GUN1 was shown to bind tetrapyrroles, Proto, MgProto, and heme via the PPR domain. In addition, GUN1 activates FC1 activity in a similar way to GUN4 enhancement of MgCh activity [104]. This model suggests a direct link between GUN1 and tetrapyrrole biosynthesis. GUN1 may regulate the distribution of tetrapyrrole biosynthesis, probably via direct interaction with enzymes [101], or through transcriptional regulation of the downstream transcription factor GLK [89]. On the contrary, it was shown that tetrapyrrole biosynthetic enzymes, such as GluTR encoded by *HEMA1*, cannot be properly imported into chloroplasts of the *gun1* mutant, which may cause impeded distribution of tetrapyrrole in the mutant [107]. On the other hand, it is proposed that binding of FC1-synthesized heme by GUN1 blocks release or propagation of the retrograde signal [104]. Currently it is unknown whether heme-binding of GUN1 is associated with ClpC-dependent degradation. It is interesting to note that heme compromises the interaction between GluTR and GluTR-binding proteins and further enhances degradation of GluTR, upon feeding ALA to *Arabidopsis* leaves [120].

It was shown that *FC1* is highly expressed in primordial tissues [68]. Mutants lacking FC1 showed poor early development with strong alleles being embryo lethal [68,69]. These results suggest FC1-derived heme functions during initial development when GUN1 is accumulated and active on proplastids to chloroplast transition. At this stage, the expression of *PhANG*s, including the key tetrapyrrole biosynthetic genes for massive Chl biosynthesis [91] remained at a low level, through PIF-dependent repression and GLK1/2 inactivation. Transcription and translation of NEP-dependent genes are also active at this stage. Considering the multiple interactions of GUN1 with proteins involved in plastid protein homeostasis and tetrapyrrole biosynthesis, it is possible that GUN1 becomes a threshold protein that may condense onto these proteins to systemically enhance these reactions (see below). In this sense, GUN1 is dispensable and its deficiency effect becomes obvious when chloroplast protein homeostasis is perturbed.

### 5.5. The Link between GUN1 and Other Processes

It has been reported that *gun1* seedling development is hypersensitive to sucrose and ABA signaling [77,121,122]. Besides, anthocyanin accumulation was differentially affected by sucrose in wild-type and *gun1* seedlings. From these results, GUN1 is proposed to have roles for sucrose and ABA signaling during initial seedling development [77,121]. Meanwhile, a yeast two-hybrid screen identified a novel GUN1-interacting protein GIP1 [111]. GIP1 was both localized to the cytosol and chloroplasts, and its abundance in chloroplasts is enhanced by NF treatment in the presence of GUN1. Although the function of GIP1 is not known, GIP1 and GUN1 may function antagonistically in the retrograde signaling pathway [111].

## 6. Outlook

### 6.1. Proposed Functions of GUN1

As discussed in this article, GUN1 functions as a biogenic retrograde signaling hub by interacting with numerous proteins (Figure 2). Although there is no experimental evidence, we hypothesize the possible function of GUN1. We propose GUN1 act as a platform to promote specific functions by bringing the interacting enzymes into the proximity of their substrates or may inhibit processes by sequestering particular pools of specific interactors [108]. One possibility is that GUN1 functions as a scaffold protein for molecular crowding that is crucial for the efficient operation of biological systems [123]. High concentrations of crowding agents entropically favor molecular association events, thereby accelerating molecular reactions [123]. Phase separation is a crucial example of when the regulation of macromolecular crowding is vital. Using genetically encoded multimeric nanoparticles (GEMs), it has been demonstrated that the mechanistic target of rapamycin complex (mTORC1) [124], the major amino acid sensor in eukaryotes [125], controls diffusion by tuning ribosome concentration. Like mTORC1, it is possible that through the N-terminal IDR and probably the PPR domain, GUN1 may form a condensate structure (droplet) that controls liquid–liquid phase separation and the biophysical properties of plastid systems involved in protein homeostasis as well as tetrapyrrole biosynthesis during initial chloroplast biogenesis (Figure 3). It is interesting to note that a particulate GUN1-derived fluorescent signal was dispersed by NF treatment [111] suggesting the relationship between functionality and droplet formation of GUN1. In this droplet, GUN1 may concentrate nucleoid for efficient NEP transcription, RNA editing, and subsequent translation of plastid-encoded proteins. Concentration of glutamyl-tRNA^Glu^ [126,127] may also be important factor for effective translation and tetrapyrrole biosynthesis. Furthermore, we cannot exclude the possibility that other signaling pathways are involved in this regulation.

The question is how GUN1-dependent droplet formation is related to the generation of the biogenic plastid-to-nucleus retrograde signal. The GUN1-deficient effect on *PhANG* derepression is only obvious when plastids become dysfunctional by NF or Lin treatment. One possibility is that the FC1-derived heme signal cannot reach the nucleus in the wild type, while it can be transferred to the nucleus in the *gun1* mutant. Considering GUN1 enhances total tetrapyrrole flow and can bind Proto and metal porphyrins, including heme and MgProto [104], we hypothesize that the GUN1 droplet controls tetrapyrrole distribution and holds the heme in the initial phase. Upon degradation of GUN1 by Clp-protease during subsequent chloroplast development, the heme is released from the droplet and transferred to the nucleus as a positive signal for *PhANG* induction. In the wild type, Clp-dependent GUN1 degradation is blocked by inhibitors [103]. Thus, produced FC1-derived heme may retain plastids in the wild type (Figure 3b). In the *gun1* mutant, certain upregulation of tetrapyrrole distribution occurs without droplet formation, which may trigger direct transfer of heme to the nucleus (Figure 3c). Meanwhile, under untreated conditions, functional plastids may produce sufficient heme that may reach the nucleus for *PhANG* induction in the wild type and *gun1* mutant. Coupled with the results from *gun2–6*, this hypothesis explains most of the observed phenotypes of the *gun* mutants.

### 6.2. Transfer of FC1-Derived Heme to Nucleus

Currently, FC1-specific heme is the most prominent candidate as chloroplast mobile biogenic signal. However, the possible involvement of other retrograde signaling pathways that interplay or antagonize the heme signaling cannot be excluded. Since hemoproteins are widely distributed within the cell, heme must be transported from the plastids to the target organelles [4]. However, compared to animals and yeast, the heme trafficking mechanism is poorly understood in plants [4]. In yeast, using a genetically encoded fluorescent heme sensor [128], it was shown that heme synthesized in the inner mitochondrial membrane can be transferred to the nucleus and the cytosol in distinct pathways [129]. Heme synthesized in the mitochondria was transferred to the nucleus via mitochondria-associated ER membrane contact sites (MCSs) that was faster than cytosolic heme transfer [129]. Although the MCSs of the chloroplast with other organelles were poorly known in plants, the ER-chloroplast MCSs were successfully visualized in a recent study [130]. Interestingly, the number of ER-chloroplast MCSs was decreased by MV treatment [130]. If FC1-derived heme is distinctly transferred to the nucleus via ER-chloroplast MCSs, rather than by the proposed cytosolic receptors, such as p22HBP/SOUL and *tau* glutathione transferases [131,132], a novel mechanism must be considered (Figure 3a).

### 6.3. A Hypothesis of the Heme Biogenic Plastid-to-Nucleus Retrograde Signaling Pathway

As shown in Figure 3a, we hypothesize the biogenic plastid-to-nucleus retrograde signaling mechanism to be: 1. At the initial phase of proplastid to chloroplast transition, putative GUN1-dependent droplets may form in proplastids, which control NEP-dependent transcription, RNA editing and translation, import of nuclear-encoded proteins, and tetrapyrrole biosynthesis. Produced FC1-derived heme may retain in the droplet. 2. During chloroplast biogenesis, GUN1 is degraded in a Clp-protease-dependent manner and the heme is released. Inhibitor treatment may disrupt the droplet formation, and GUN1 remains undegraded by preventing Clp-protease activity. 3. FC1-derived heme bound to GUN1 droplets is released and emitted from plastids via cytosolic or an ER-plastid MCS-dependent pathway. 4. In the nucleus, heme may inactivate PIFs or activate transcription factors, GLK1, and/or HY5. 5. Consequently, *PhANG* expression is activated. 

At present, it is unknown whether FC1-derived heme affects the chloroplast processes, such as transcription, editing, translation, and import (Figure 2). Previous proteomic analysis of heme-binding protein detected none of the GUN1-interacting proteins in *Arabidopsis* [72]. Therefore, further study is required to elucidate this possibility.

### 6.4. Perspectives

It is possible that such a GUN1-dependent droplet functions as a system, so it will be difficult to dissect the individual pathways. To clarify this hypothesis, in addition to biochemical and molecular biological methods, novel approaches to liquid–liquid phase separation would be beneficial. For instance, live imaging using GEMS [124] and labeling methods identifying proximal and interacting proteins [133,134] may be successful. The future challenge will also be to identify the components of the signaling pathway that link heme to *PhANG* expression. Live cell imaging using the fluorescent heme sensor [128] would aid in the understanding of this mechanism.

## Figures and Tables

**Figure 1 plants-10-00196-f001:**
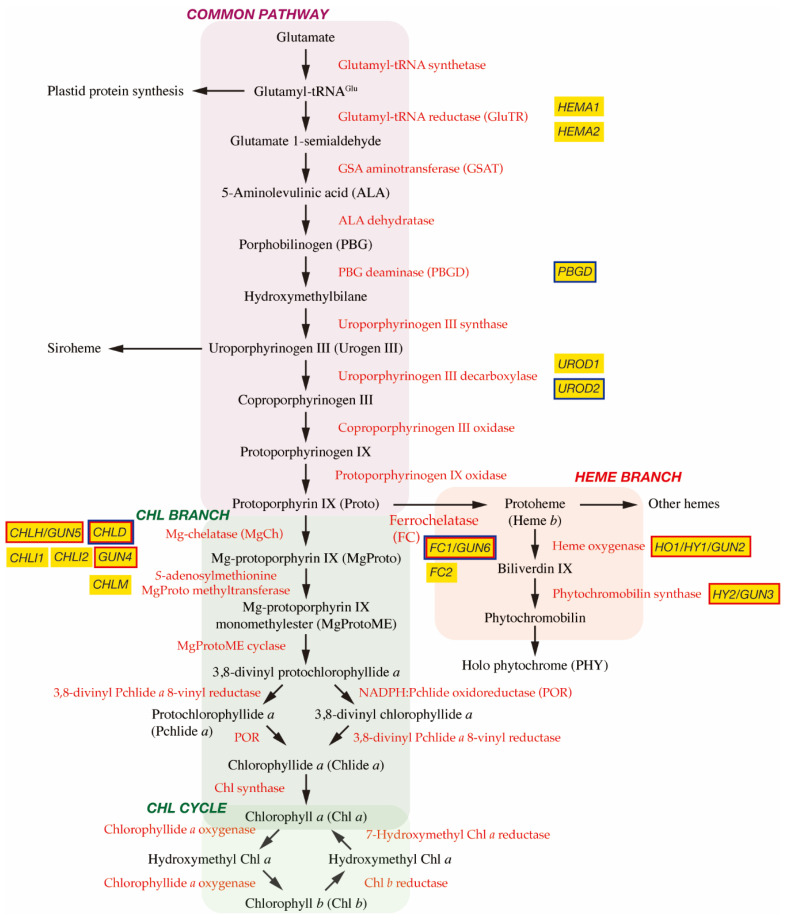
Tetrapyrrole biosynthetic pathway. Enzymes involved in the tetrapyrrole biosynthetic pathway are indicated by red. Important genes described in this article are shown in yellow boxes and genes encoding GUN proteins are indicated by red borders. It is noted that *GUN2*~*GUN6* genes are found at the branch points of Chl and heme biosynthesis. GUN1 interacting proteins are indicated by blue borders.

**Figure 2 plants-10-00196-f002:**
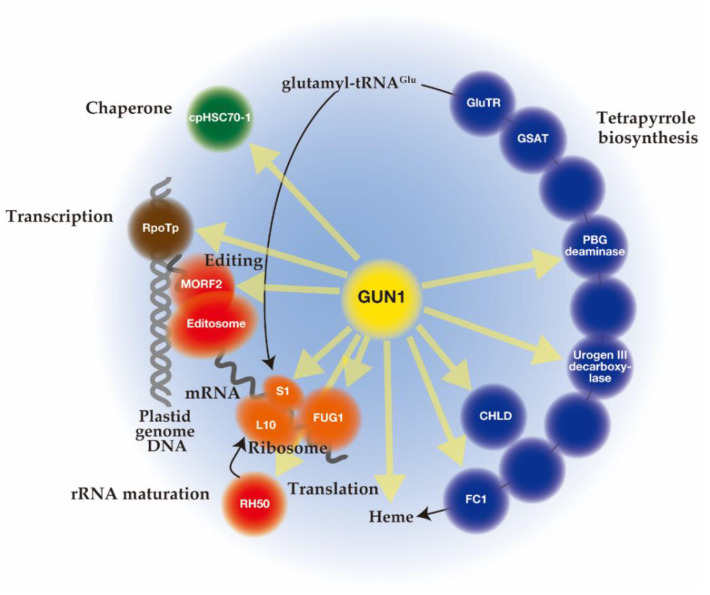
Schematic overview of genomes uncoupled (GUN)1 interacting proteins involved in plastid protein homeostasis (transcription (brown circles), editing and maturation (red circles), translation (orange circles), and chaperone (green circles)) and tetrapyrrole biosynthesis (blue circles). Yellow arrows indicate GUN1 interactions. It is possible that through the N-terminal intrinsically disordered region (IDR) region or pentatricopeptide repeat (PPR) domain, GUN1 forms a droplet that causes molecular crowding, which enhances entropically favor molecular association events, thereby accelerating molecular reactions. Interactions of GUN1 with indicated proteins were demonstrated by co-immunoprecipitation, bimolecular fluorescence complementation (BiFC), and yeast two-hybrid assays.

**Figure 3 plants-10-00196-f003:**
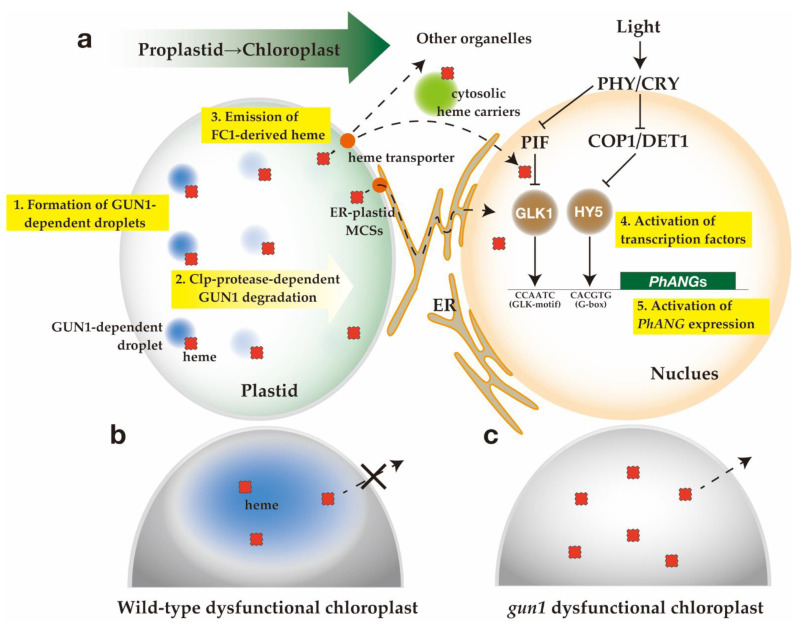
A hypothesis of the heme biogenic plastid-to-nucleus retrograde signaling pathway. (**a**) In plastids (left), the transition of proplastid to chloroplast is indicated from left to right. Blue circles indicate GUN1-dependent droplets and red symbols represent ferrochelatase (FC)1-dependent heme. 1. At the initial phase of proplastid to chloroplast transition, a putative GUN-dependent droplet may form in proplastids, which enhances nuclear-encoded polymerase (NEP)-dependent transcription and translation, import of nuclear-encoded proteins, and heme biosynthesis. Synthesized heme may bind to the GUN1-dependent droplets. 2. During chloroplast biogenesis, GUN1 is degraded by Clp-protease-dependent manner, causing disappearance of the blue circles. 3. FC1-derived hemes bound to GUN1 condensates were released and emitted from plastids via cytosolic or ER-plastid membrane contact sites (MCSs)-dependent pathway (dashed lines) through a plastid-envelop-localized putative transporter (orange circles). In the cytosolic pathway, heme may bind to cytosolic heme carrier proteins (green circles) to reach other organelles. 4. In the nucleus, heme may inactivate PIFs or activate transcription factors, GLK1, and/or HY5. 5. Consequently, *PhANG* expression is activated. (**b**) In the wild type, when chloroplasts are rendered dysfunctional by inhibitor treatments, Clp-protease-dependent degradation is protected resulting in the dispersion of the GUN1 droplet (blue circle), as well as retention of heme (red symbols) in plastids. (**c**) In *gun1*, when the chloroplasts become dysfunctional, heme can be emitted from plastids because the GUN1-dependent condensates are deficient.

## Data Availability

Data is contained within the article.

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
