# Peer review of "The Role of Tetrapyrrole- and GUN1-Dependent Signaling on Chloroplast Biogenesis"

_plants, 2021, doi:10.3390/plants10020196_

Round 1

Reviewer 1 Report

The molecular mechanism behind the plastid-to-nuclear retrograde signalling is one of the hottest topics attracting humorous attention from global researchers. This review emphasizes the tetrapyrrole mediated retrograde signaling in plants and algae and summarise recent advances in this topic. From a historical view, the authors gave a nice survey on the development of this topic, too. I have several points to improve this review.  

Line 32: while chl a and b are found in plants, additional chl derivatives such as chl d, f exist in algae. could authors check this point? 

Line 83, authors need to rearrange the sequence of references 46 and 47, which should be listed as references no. 17 and 18 in the manuscript. More importantly, references 46 and 47 did not study differential tisse-specific and developmental expression profiles of FC1 and FC2. 

The heme-binding capacity of TSPO might be discussed in the review (Vanhee et al., Plant Cell,2011). It is not necessary to use the abbreviation of TSPO as it is mentioned once in the manuscript. 

Reference 54 in line 217 needs to be replaced by Youbayashi et al., Plant Cell Physiol (2016).

Lines 255-269, Arabidopsis contains two homologs of GLK, and GLK1/2 showed overlapping function in the regulation of photosynthesis. I suggest using GLK1/2 to replace GLK1.

Reference 94 has shown GUN1 is able to bind heme. It remains questionable whether or not heme-binding of GUN1 is associated with ClpC-dependent degradation of GUN1. Recently, it has been shown that heme compromises the interaction between GluTR and GluTR-binding proteins and further enhances degradation of GluTR, in particular upon ALA feeling of Arabidopsis leaves (Richter et al., elife, 2019).  

Reviewer 2 Report

Dear authors,

I have read your review titled “The role of tetrapyrrole signaling on chloroplast biogenesis” with great interest. The review while focusing on the role of tetrapyrrole signaling on chloroplast biogenesis presents a clear and thorough overview on tetrapyrrole biosynthesis, biogenic retrograde signals, and the potential role of GUN1 in these processes. Moreover the review includes new hypothesis for the action of GUN1 that could be explored in future works contributing to the advance of the field. Due to the recent interest and increasing number of publications on the role of plastid retrograde signals, the review is of great interest for researchers on the field. The review is nicely written and organized, and the figures are clear and informative. I only have some comments and suggestions to help optimize the manuscript before publication.

Regarding the main text there are some major issues that should be corrected:

  • From line 463 onwards: It should be stated more clearly that the following are hypotheses of the authors that do not have any experimental support yet. Also the potential involvement of other potential signals cannot be dismissed and should be mentioned, even briefly, in this section. Title of figure 3 should reflect this so I suggest: “A hypothesis of the heme biogenic plastid-to-nucleus retrograde signaling pathway”
  • Line 83: Check references and add the correct ones, since 46 and 47are not about FC.
  • Line 170: This statement is not supported by later research. It was done using the single mutant, and as stated above (line 162 Huang et al 2009) the double chli1/chli2 has gun phenotype in NF.
  • Line 220-221: the use of “confirming” in this sentence is not backed by the results presented in the following sentences. I suggest changing it for “supporting” or similar.
  • Line 282: It is not clear what the implications of “turn disordered to the ordered structure upon binding” could be from this sentence. Does it involve GUN1 stability as stated in reference 94, or as it is explained in figure 2 legend “forms a droplet that causes molecular crowding”? Please clarify.
  • Line 293 (and 406): GUN1 has not been reported in meristems. Ref 92 "newly emerging first true leaf at the shoot apical meristem now starts to express GUN1" so not in the meristem but the leaf primordia or developing leaves. Please change these sentences.
  • Line 297-300: This is not the only reason why it is believed that GUN1 functions only in the biogenic retrograde signals. The phenotypes of the mutants are seen only at early stages of development, and in cotyledons. Reference 92 could be included as they authors describe a potential role in flowering.
  • Line 345: FUG1 alias is cpIF2 (not Clp2) for chloroplast translation initiation factor IF-2.
  • Line 355. Include reference 90 (Tadini et al. Plant Physiol 2016) for GUN1 interactors.
  • Line 394-395: As it is the sentence is a repetition of the sentence above. Please clarify the conditions for Pchlide accumulation in gun1 in reference 70 or merge both sentences and references.
  • Line 406: Rephrase this sentence since GUN1 is not regulated at transcriptional level but at protein level (paragraph starting in line 291) and has not been described to be stabilized in meristems but emerging leaves.
  • Line 465: GUN1 does not suppress tetrapyrrole flow totally (e.g. gun1 mutant accumulates Pchlide), and can also bind MgProto. Please rephrase.
  • Line 469: The sentence as it is is not clear. Do the authors mean that inhibitors of chloroplast development block GUN1 degradation and the droplet formation is affected? Please clarify.
  • Figure 1. In the Chl branch, the step from 3,8-divinyl protochlorophyllide and Chlorophyllide is divided in two. This is not explained in the main text (line 72 to 74) or the legend of the figure. I would suggest leaving the main pathway in the figure, or else explain it in the main text.
  • Figure 2. Interaction between GUN1 and CHLD has been reported (Y2H and co-immunoprecipitation) and should be included in the text and the figure (reference 90)
  • Figure 2 legend. Add the description for arrows, and circles of different colours.
  • Figure 3 Please include what are the red squares (heme?) and associated blue circles (GUN1? and the fading is degradation of GUN1?), red and blue circles in the plastid membrane, and meaning of the different arrows.

Here are some comments and suggestions:

  • Line 2: Title. GUN1 signaling has a prominent role in the abstract and comprises more than half of the manuscript. I suggest its inclusion in the title.
  • Line 14: There is not a clear molecular function identified yet for GUN1, just interactions and working hypothesis, so I suggest rephrasing this sentence.
  • Line 47-49: reference 9 is form cucumber while the sentence refers clearly to Arabidopsis, remove the reference or specify the species for each statement.
  • Line 52: should the name inside the brackets be UROGEN III like in figure 1?
  • Line 71: Inside the brackets MgProto-Me
  • Line 89 to 95: This paragraph lacks references. Also, since the information on plastids biogenesis is not extensive some references on plastid biogenesis could be added as it is done in line 113 for retrograde signals. A suggestion: Pogson BJ, Ganguly D, Albrecht-Borth V (2015) Insights into chloroplast biogenesis and development. Biochim Biophys Acta 1847: 1016–1024
  • Line 234: The models in the references were published before 2019, when exhaustive analyses in references 65 and 68 were reported. More recent reviews have removed ABI4 and PTM from the models andcan be mentioned. E.g. Hernández-Verdeja T, Vuorijoki L, Strand Å. Emerging from the darkness: interplay between light and plastid signaling during chloroplast biogenesis. Physiol Plant. 2020 Jul;169(3):397-406. doi: 10.1111/ppl.13100.
  • Line 242: Please include all references 65-68, 71
  • Line 255: Please refer to both GLK1 and GLK2 transcription factors at the beginning, since they can complement each other, and make it clear in the following text which one is prevalent.
  • Line 262 to 266: The information provided is not clearly explained or relevant. Some of the references are from plastid development in roots (82, 83) and this should be clearly stated in the text, as it is not a natural system for chloroplast development. Please rephrase.
  • Line 321-325: This paragraph lacks references and it would help the interested reader to have some references on plastid transcription. Suggestions:

Pfannschmidt T, Blanvillain R, Merendino L, Courtois F, Chevalier F, Liebers M, Grübler B, Hommel E, Lerbs-Mache S. Plastid RNA polymerases: orchestration of enzymes with different evolutionary origins controls chloroplast biogenesis during the plant life cycle. J Exp Bot. 2015 Dec;66(22):6957-73. doi: 10.1093/jxb/erv415. Epub 2015 Sep 9. PMID: 26355147.

Liebers M, Grübler B, Chevalier F, Lerbs-Mache S, Merendino L, Blanvillain R, Pfannschmidt T. Regulatory Shifts in Plastid Transcription Play a Key Role in Morphological Conversions of Plastids during Plant Development. Front Plant Sci. 2017 Jan 19;8:23. doi: 10.3389/fpls.2017.00023. PMID: 28154576; PMCID: PMC5243808.

  • Line 420: There is another recent reference for GUN1 involvement in anthocyanins accumulation that should be included: Richter AS, Tohge T, Fernie AR, Grimm B. 2020 The genomes uncoupled- dependent signalling pathway coordinates plastid biogenesis with the synthesis of anthocyanins. Phil. Trans. R. Soc. B 375: 20190403. http://dx.doi.org/10.1098/rstb.2019.0403
  • Line 498: Steps 4 and 5 of figure 3 should be mentioned here as has been done with 1 to 3.
  • Figure 1. The name of the different branches of the pathway could be highlighted is some way (maybe uppercase, or bold). As it is, it is difficult to find the name of the branch and link it to the box enclosing it. Could the boxes of the CHL subunits be clearly grouped by the name of the enzyme? GUN5 and GUN4 are one step above the pathway and can be misleading.

Minor language, spelling and typos:

  • Line 44: “The” following two enzymes
  • Line 83: protoheme “is” further
  • Line 154: gun2 and gun3 “mutants are” unable
  • Line 162: shown to “have” a gun phenotype
  • Line 186: Arabidopsis “seedlings”
  • Line 331: and “regulate” RNA
  • Line 409, 462, 474: PhANGs refers to genes and should be in italics.
  • Line 465: can “bind”
  • Line 496: preventing Clp-protease “activity” (?)

Reviewer 3 Report

MS ID#: plants-1057547

MS TITLE: The Role of Tetrapyrrole Signaling on Chloroplast Biogenesis

Dear Editor, Dear Authors,

Thank you for letting me read this interesting work on the role of tetrapyrroles as signaling molecules. Authors are experts in the field and the work will be a significant contribution to the retrograde signaling field, by resuming recent advancements and showing future directions of the research.

At the same time, it is my opinion that the MS should undergo a revision before being acceptable for publication.  Here my major remarks:

  • The title is on Tetrapyrrole and chloroplast biogenesis, but the MS lacks a clear description of what “chloroplast biogenesis” is. I would include few lines describing this physiological process and the main steps. Also, I would report some names of the pathways related to operative retrograde.
  • Being this MS a Review, I would homogenize the use of citations. There are many sentences, which would require a reference, with none and other sentences showing several references.

Example:

“In plant cells, the chloroplast is one of the differentiated states in which plastids have a photosynthetic function. In the meristem of angiosperms, plastids exist as undifferentiated proplastids, and chloroplasts can directly form from the proplastids with developmental cues and light signals. In the absence of light, proplastids differentiate into etioplasts with unique lattice membrane structures called prolamellar bodies (PLBs), which accumulate Pchlide a and POR. Once the etiolated seedlings are exposed to light, most Pchlide a molecules in PLBs are immediately converted to Chlide a by POR and then to Chl a via enzymatic processes.

Plastids originate from a free-living cyanobacterium in a process known as endosymbiosis [18].

  • I would adopt a less-cryptic English style. The aim of reviews is to collect recent advances in the field, but also to make them enjoyable by a larger public. In my opinion, authors should make an effort in choosing direct wordings and making the sentences more readable.

Example:

“Interestingly, GUN1 contributes to various biological processes, through processes such as transcription, translation, and protein import.” It would be more direct to explicit which processes GUN1 is involved in and use by regulating instead of “through processes”.

  • In my opinion the MS should undergo professional English proofreading.

Example:

“The highly disordered region, which may correspond to the intrinsically disordered region (IDR) [93], is found in the N-terminal region of GUN1 [94], which will turn disordered to the ordered structure upon binding with partner proteins”

  • In my opinion several molecules act together or in distinct situations as retrograde signals. These results have been obtained by many research groups working on PAP, MEcPP and beta-cyclocitral, among others. In the text there are at least two affirmations that put tetrapyrrole as the main and only thrustable signal. Is this sustained by scientific literature?

Detailed suggestions:

Line 2: Chloroplast biogenesis is in the title, but a description of this event lacks in the MS

Line 16: through processes -> by regulating, modifying, altering

Line 24: Reference

Line 94: Reference

Line 105: nucleus does not usually count as organelle

Line 107: nucleus controls most …. Reference

Lines 110-113: This is the distinction between biogenic and operational. This is the only description of the pathway. Why should a treatment with lincomycin or norflurazon be classified as biogenic rather than operational? Please expand the explication.

Line 113: Complicated. Pathways may be complex, multifactorial, hard to decode. Complicated is not very scientific, especially in a review.

These refers to all the pathways or only to operational retrograde signaling?

Line 117: Reference

Line 127: which/that encode

Line 130: several proposed molecules are solid retrograde signals, such as Beta cyclocitral, MEcPP and PAP. Even if the review is focused on tetrapyrroles authors should avoid diminishing other known pathways

Lines 131-137: were performed? Not clear “for gun phenotype evalutation”

Line 148: complicated. As above

Line 154: unable. Make unable?

Line 156: Leaded. Led?

Line 162-163: of MgCh? Mutants cannot be a phenotype.

Line 164: Were-> Was; treatment of ->With

Line 165: The increase in fluorescence was observed only after ala treat…? Also? In agreement?

Line 168: acts. I suggest to be direct a say what means “negative regulation”. Decreased expression? Accumulation?

Line 171: a* gun phenotype

Line 180: has been proposed

Lines 184-187: please add more detail of the RNAi system used.

Line 191: certain has several meanings.

Lines 194-196: Rapidly degraded. Containing possessing. Not clear.

Lines 201-202. Please extend the sentence. An alternative hypothesis is that another metabolite, such as protoheme, functions as ….

Line 2010-211: I do not understand the sentence. A flux? Specific? In this case explain why specific.

Line 212: “the” gun phenotype

Line 213: FC1-producing heme?  I would use derived, dependent, produced.

Lines 211-214 as a signaling as a retrograde. Double

Line 215: has? Who?

Lines 216-218: Please add a more detailed explanation.

Line 221: as above (producing)

Line 225: please name same of the proteins

Line 230-242: ABI4-PTM results have been challenged. Please cut all the explanations and citations of what authors think is no longer valuable and directly jump to the conclusion: However…. 

Line 271: gun2-6 looks like an allele 2.6. Maybe saying “mutant lines gun 2 to 6 would be easier”.

Lines 281-283: This sentence need revision

Line 299: has been suggested

Line 301: suggested -> hypothesized, reported, observed…

Line 303: a protein is an element/unit rather than a part of a complex.

Line 310: plant developmental stage

Line 310-311: functionality of Gun1 when Gun1 function is different? Please be less vague

Line 325: reference

Lines 352-375: When authors want to compare two interpretations present in literature, they should state that in advance rather than explaining the first and abruptly introducing the second just after.

Line 428: The chapter 5.6 and 6 are very similar. Consider fusing them.

Line 477: reference. In my opinion there are many metabolites acting as retrograde signals together or in distinct situations.

Comments on the model:

GUN1 accumulates in droplets and keeps heme from going to the nucleus and sustain Phangs expression. How can a retained retrograde signal affect chloroplast processes and chloroplast import? As author’s model is the key part of the article, I would prefer an extensive explanation. Figure 2 and the targets of GUN1 could be recalled here.

In my opinion the title should really be at the center of this model, therefore Tetrapyrrole accumulation and chloroplast biogenesis. Conversely, the model is on GUN1 and do not really include the physiology of chloroplast biogenesis.

Regarding Figure 3: The droplets and their degradation is not clear. Maybe a darker color or a border could help. If heme arrives in the nucleus, it would be worth to draw it also there. Maybe the size of heme could be reduced.

Round 2

Reviewer 3 Report

I think the manuscript ID “plants-1057547” greatly improved after the first revision and represent now a truly significant contribution to the field.

Minor comments:

Lines 36-39: I would add “on other aspects of plant physiology.” At the end of the sentence to contrast with this review on the role on chloroplast biogenesis.

Line 71 and 74: put the “a” immediately after the molecule and report the short name after: “3,8-divinyl (DV) protochlorophyllide 70 (Pchlide) a.” should be “3,8-divinyl (DV) protochlorophyllide 70 a (DV-Pchlide a).”

Line 94: by red borders.

Line 112-114: the sentence is not clear. Photosynthesis is not a reaction with light, rather a reaction that uses/captures/exploits light energy that takes place in the photo…. At the level of thylakoids…. Separate from Calvin cycle.

Line 120: Do retrograde signals control gene expression? I would use alter, modify because the exact mechanism is not really described.

Line 163: Singlet Oxygen

Line 183: the verb “was detected” is better at the end of the sentence

Line 183: A treatment with Mg …

Line 204: in the gun5 background….

Line 205: Significantly decreased….

Line 206: in control conditions or NF treated?

Line 209: ring structure, which has photo….

Line 284: As a matter of fact, / Indeed the overexpression…

Line 298-302: may I suggest rephrasing for simplicity as follow: The highly disordered domain at the N-terminus of GUN1 may correspond to an intrinsically disordered region (IDR). The binding of protein partners induces a conversion of this domain to an ordered structure, which allows....

Line 309: “It was demonstrated that” is not necessary. References are there to show that it was demonstrated.

Line 335: Yellow arrows indicate Gun1 interactions.

Line 457: there is no experimental

Line 458: Hypothesize or propose not both of them

Lines 463-466: all this could become a “similar to mTORC1 [124].” The sentence is not sufficiently explained otherwise, and the animal TOR is abruptly introduced.

 Line 512: “As a mobile signal of biogenic plastid-to-nucleus signaling on chloroplast biogenesis, currently”

Currently, FC1-specific heme is the most prominent candidate as chloroplast mobile biogenic signal.
